# Analysis of Defects in Residential Buildings Reported during the Warranty Period

**Edyta Plebankiewicz \* and Jarosław Malara** 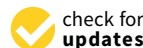

Cracow University of Technology, Warszawska 24, 31-155 Kraków, Poland; jmalara@l7.pk.edu.pl

\* Correspondence: eplebank@L7.pk.edu.pl; Tel.: +48-12-628-2330

**Abstract:** The aim of the article is to present the results of preliminary research of the defects in residential buildings occurring during the warranty period. Due to the small amount of data, the research results cannot be generalized but allow for the formulation of research hypotheses that will be verified in future studies. The data collected included reports of defects in three multifamily residential buildings constructed by the developer in one of the big cities in Poland, which were then examined. For the examination of defects, statistical analysis was used, which revealed that more than half of the reports contained reasonable defects. The results of the preliminary research also indicate that, on the one hand, owners are very active in making warranty claims in the first three months from the date of commissioning, and, on the other hand, with time, the percentage of reasonable defects increases. In terms of the significance of defects, the largest percentage was significant defects. The results showed little activity on the part of property managers in the initial phase of the operation of the buildings, which is the opposite of that of apartment owners. Reports of faults in windows and door joinery, moisture, scratches on walls, and in the area of balconies and terraces are characterized by a relatively low number of cases reported in the first half of the year after the building is commissioned and a gradual increase in the subsequent warranty period. On the other hand, reports related to electrical installation defects are most frequent in the initial period of the warranty, but, with time, their number decreases.

**Keywords:** residential buildings; warranty; acceptance of the work

## 1. Introduction

By concluding a contract of works, the contractor undertakes to hand over the facility provided for in the contract and the contracting authority to pay a specified remuneration. The key stage in mutual relations is the acceptance of works. However, the obligations of the parties do not end with the acceptance. Under legal regulations, as well as the usual relevant contractual provisions, the contractor takes responsibility for the defectiveness and noncompliance of the performed works with the contract. In the Polish legal environment, the rules of liability are determined by guarantee and warranty. These two types of liability of the contractor operate independently of each other, showing both similarities and significant differences. The main difference is that granting the guarantee is a voluntary declaration by the contractor to assume responsibility for defects in the works performed, which is regulated in the contract. On the other hand, the warranty is legally binding and is regulated in detail in the provisions of the Civil Code [1].

According to Article 556 of the Civil Code, the warranty is an inalienable right of the buyer, while the seller is liable to the buyer if the item sold has a physical or legal defect. A physical defect is defined as one of four possible situations concerning the sold item:

1. There are no features that such a thing should have in view of the purpose in the contract, as indicated or resulting from the circumstances or purpose.

2.　　There are no features that the seller has provided the buyer with, including the presentation of a sample or pattern.
3.　　Not fit for the purpose of which the buyer informed the seller at the conclusion of the contract, and the seller did not object to such a purpose.
4.　　The item is issued to the buyer in an incomplete state.

However, when identifying potential defects, it is important to keep in mind the limited time for reporting them. In the case of a property defect, it is a period of five years from the date of handing it over to the buyer, which, with regard to construction works, takes place at the time of acceptance of the works [2,3]. It should be noted that the detailed specifications of the contractor's liability are described in the national legislation of individual countries. While the paper refers to Polish regulations, the laws of other countries also include the concept of warranty, differing in only some formal and procedural details.

In the context of the warranty, the provisions of the Civil Code do not describe the fault but the defect of the item sold. In this paper, the notions of fault and defect are treated as equivalent due to the way the issue is described in Polish publications, where these terms are also interchangeable.

Defects are a common phenomenon in the construction industry in all countries. Contractors, as well as investors, should pay special attention to them as they can have a significant impact on the costs and required resources of the project. In the literature, there are many publications related to the frequency of defects in residential buildings conducted in various countries [4–26]. However, the vast majority of the publications are related to defects reported during acceptance. The research of the defects in buildings occurring during the warranty period may become an important contribution to the analysis of defects. The aim of the article is to present the results of preliminary research of the defects in residential buildings occurring during the warranty period. Due to the small amount of data, the research results cannot be generalized but allow for the formulation of research hypotheses that will be verified in future studies.

This paper is organized as follows. Section 2 contains the literature review. Section 3 introduces formal and legal conditions relating to defects reported during the warranty period. Section 4 describes the analysis method and results. Finally, Section 5 presents conclusions.

## 2. Literature Review

The acceptance of construction works is regulated differently in the legal systems of each country. For example, as follows from [5], in the German legal system, the subject of acceptance of construction works is regulated in more detail than in the Polish legal system. However, a certain fundamental similarity can be seen in all systems, which is the similarity of the effects of acceptance of construction works, including, above all, the payment of the contractor's remuneration and the calculation of the limitation period for claims concerning the liability for defects.

Defects are a common phenomenon in the construction industry. In the literature, there are many publications related to the frequency of defects in residential buildings. In the years 2009–2012, Ojo and Ijatuyi [6] conducted research into defects on the example of the Sunshine Gardens housing estate in Akure, Nigeria. The most common defects found in the roof structure and covering included the use of improper quality materials, improper wood treatment, poor quality of workmanship, and inaccurate supervision of construction workers. The walls of the analyzed buildings were made in the wrong way, low-quality materials were used, and short window and door lintels were applied. Moreover, the floors were made of low-quality materials. Rotimi et al. [7] presented the example of housing construction in New Zealand to determine the level of detection of defects by independent building inspectors at the time of handing over 216 new residential buildings in the years 2008–2011 and examined the number and types of defects found. The most common defects included uneven painting, nail marks, poor quality of room and floor finishes, incorrectly fixed handles in doors and windows, cracks in buildings, and incorrect installation of toilets. In [8], the results of a research project analyzing the location and type of 3647 faults located in 68 residential buildings in Spain

are presented. The research showed that the most frequent defects were inappropriate materials or components, their bad location, surface defects, including uneven surfaces, scratches, cavities, faults of machines, and components affecting their functionality, such as doors rubbing against the floor or nonfunctioning air conditioning. Shirkavand et al. [9] presented the numbers of defects detected during acceptance in Norwegian construction projects, mainly in Trondheim. Seven buildings were analyzed: a kindergarten, four nursing homes, a school, and apartments. The most frequent defects were damage to surfaces and installation networks. Ismail et al. [10] performed a study to investigate the most common defects in 72 new terrace houses in Malaysia. The most common defects were ones on the corners of walls, uneven joints, lack of angles and planes of walls, unevenly painted walls, cavities in wall tiles, doors and windows not closing, unfinished works (such as the unfinished installation of railings), and dampness. The results of studies concerning the number and type of defects are also presented in [3,11–14].

A number of studies have also been devoted to the causes of defects. Ahzahar et al. [15] investigated the factors contributing to construction failures and defects in Malaysia. According to the authors, defects and faults in buildings are affected by, among others, certain building materials, errors during construction, corruption, lack of supervision, and design errors. Mesa Fernández et al. [16] analyzed various factors in quality control in residential construction projects in Spain. The authors proposed changes in control, for example, focusing not only on the management process but also on product quality and greater control of material deliveries. In [17], the defects reported by the representatives of the cooperative are presented and analyzed, focusing on the relationship between the characteristics of the building, the size of the developer/provider company, and the type of defect. According to the authors, the size of the developer company and the location of the building have a significant impact on construction defects. In [18], the main factors influencing the occurrence of defects at the design stage of residential buildings in the Gaza Strip are identified and ranked. For the purpose of analysis, a survey was conducted, which indicated three main design errors: ignoring or incorrectly performing the ground survey, lack of qualified supervision over the drawings, and conflicts between architectural and structural drawings.

On the basis of the literature study, it can be concluded that the main causes of construction defects are poor quality of materials, poor quality of workmanship, and inaccurate control of construction works. During acceptance, defects most often appear on walls: the greatest detection of defects on surfaces may result from the fact that these defects can be easily found visually, and no specialist equipment or great effort is required to detect them.

The purpose of the research on defects in housing construction is aimed at not only quantifying the defects and identifying their causes but also at estimating the costs to be incurred for the necessary repairs. Based on the research conducted by Mills et al. [19] in Australia from 1982 to 1997, it can be concluded that the cost of repairing the defects constitutes around 4% of the contract value. On the other hand, Josephson [20] concluded that the cost of defects corresponds to 4.4% of the construction costs of buildings, and the time to repair them amounts to 7% of the total working time. Kucharska-Stasiak and Mielczarek [21] studied the quality of housing construction on the example of Widzew EF housing estate in Łódź from 1977 to 1980. The authors calculated the costs of correction works on the basis of the collected data and estimated the losses incurred by the contractor due to the poor quality of workmanship.

Building defects are the main reason for exceeding the budget of a construction project, hence the search for a model approach to defect management. Park et al. [22] are working on a construction defect management system involving augmented reality (AR) and modeling information about the BIM (Building Information Modeling). In [23], a model for forecasting defects of multistory reinforced concrete buildings is presented using neural networks (NN–PSO classifier), while in [24], the causes of defects and faults are analyzed using trees and risk measures. In [25], a model based on LDA (Loss Distribution Approach) is presented for the assessment of the system of responsibility for repairing defects and faults in the period of responsibility for defects in residential buildings using the LDA

loss distribution method. In turn, Oswald and Abel [4] discussed the procedure for the assessment of defects found in buildings. The authors created and described a block diagram for the evaluation of optical defects and presented a method of utility analysis consisting of considering several alternatives in terms of criteria and selection of the best solution and the Aurnhammer's method for determining the reduction of value when defects are found.

## 3. Formal and Legal Conditions

The buildings analyzed in the article were implemented by the developer. The contract between the developer and the buyer of an apartment, in accordance with Polish law, is governed by the Act on Protection of Buyers of Apartments or Single-Family Houses [4–27]. According to the Act, the transfer of ownership to the purchaser is preceded by acceptance in the presence of the customer or his proxy (notarial power of attorney is required). Before taking over the residential unit, the developer is obliged to notify the buyer of this fact. Based on the legal status, as well as court decisions, the authors have prepared a graph depicting the moment of commencement of the warranty period in the case of developer investments (Figure 1).

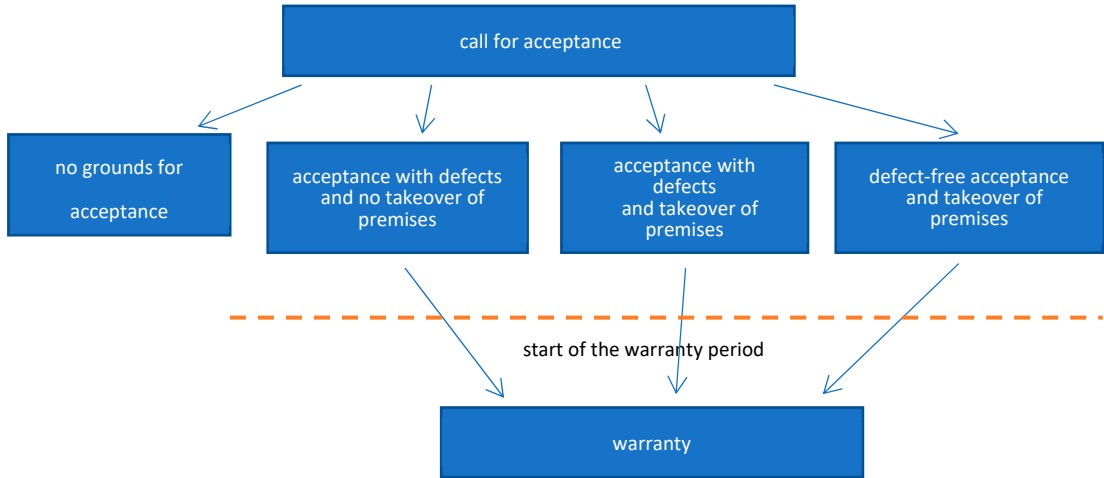

**Figure 1.** Diagram showing the start of the warranty after acceptance.

The buildings analyzed in the paper were executed by a developer. According to the law, the transfer of ownership to the buyer is preceded by acceptance in the presence of the customer. During the acceptance meeting, the client or the company supporting the acceptance inspects the apartment, after which an acceptance protocol is to be prepared, to which the buyer or their proxy may report any defects or faults found during the acceptance meeting. The customer, as the receiving party, has the right to decide which defects will be included in the acceptance protocol. He or she has the right to report all objections to the apartment, even the least important ones, such as dirt on the handles.

After the possible identification of minor defects and preparing the acceptance report, the apartment is handed over to the client. Before handing over the apartment, minor assembly work is usually done. In case of detecting significant defects and faults, the apartment is not handed over to the client, and the developer is obliged to repair the defects. At the time of acceptance, the customer agrees to the condition of the apartment. This means that if the customer has not noticed any defects during acceptance, or he or she has not decided to put a defect noticed in the acceptance report, the customer has no right to demand its repair. If the defect was, for example, hidden or a result from the operation of the building, the buyer of the apartment can take advantage of a 5-year warranty for construction works because, during this time, the developer is obliged to maintain the so-called "acceptance status" of the flat.

To a large extent, the results presented in this paper form a continuation of the research on defects occurring at the acceptance stage [3,27]. The research results presented here involve the case of a couple of buildings, one of which had a statistical analysis of defects performed at the acceptance stage.

## 4. Methods

### 4.1. Data Collection

The results presented in this paper are based on data that were collected through analysis of the reports on the state of defects made by the inspector during the warranty period. The area of the conducted research included a total of 3 residential buildings. The buildings are part of various projects but were produced by the same developer and contractor. Two of them possess 16 aboveground stories, which are divided into 3 staircases with 172 flats, each with areas ranging from 29 to 117 m$^2$. These buildings were accepted in 2017 and 2018, which, in further analysis, coincides with the beginning of the research. The third building is a 6-story one, with two staircases and 73 flats. The time of the building's acceptance was August 2018. Details of the facilities analyzed are presented in Table 1.

**Table 1.** Details of the residential buildings.

| Building | The Number of Stories | The Number of Staircases | The Number of Flats | Total Area of Flats (m$^2$) | The Time of Acceptance |
|---|---|---|---|---|---|
| A | 16 | 3 | 172 | above 8000 | 2017/2018 |
| B | 16 | 3 | 172 | above 8000 | 2017/2018 |
| C | 6 | 2 | 73 | 3000 | August 2018 |
| Total | | | 432 | 19,000 | |

All buildings were finished to the same degree. The differences were in the type of finishing materials used in the common areas (that is, in staircases and the outdoor area). The development standard included plastering inside the premises, walls prepainted with white paint, screed, window and balcony joinery, entrance doors, embedded internal and external window sills, distributed ventilation system with diffusers for mechanical ventilation, distributed central heating installation together with installed radiators, water and sewage system without fittings, electrical installation together with plug and lighting sockets and switchboard, distributed teletechnical installation with a collective box, sockets and intercom, balustrades, and tiles on external floors. A precise definition of the elements of the shell unit standard is important in terms of warranty claims.

The collected results of the research present an assessment of the validity of defect reports made by customers. The data are the result of actions taken by the developer, the supervision inspector, and the contractor. Each defect included in the statistics underwent verification of the original position of the developer. The role of the supervision inspector is to make an independent assessment and qualification of the report of potential defects by customers and managers of individual properties. The results of the research presented in this article are, therefore, based on expert knowledge.

The collected data on warranty notifications cover the period from building acceptance in January 2018 and August 2018 to March 2020. A total of 560 notifications on the occurrence of defects during the warranty have been identified in the course of the conducted research. The collected data were analyzed statistically using GNU PSPP Statistical Analysis Software. First, the authors determined the percentage of valid and unfounded claims and, next, the percentage of claims under warranty cumulatively. Subsequently, the statistics of the three-stage qualification of defects were established—of low, medium, and high significance—as well as the relationship of defects to the place where they occur. The successive analyses concerned the type of defects. Pearson's parametric correlation and the determination factor R$^2$ were computed to assess the relationship between the defects reported in different buildings and to establish the strength of the interdependence between different groups of defects.

*4.2. Data Analysis*

A total of 560 notifications on the occurrence of defects during the warranty were identified. The research concerned 353 defects qualified as valid, which accounted for 63.04% of all reports. Additionally, 207 defects were considered unfounded, which accounted for 36.96% of the reports. Table 2 presents the summary statistics of the reported defects. Buildings A and B are larger both in terms of the number of floors and the number of flats. Moreover, they were accepted earlier than Building C, which significantly affected the number of reports identified in the study.

**Table 2.** The statistics of defects.

| Building | Number of Defects | Valid | | Unfounded | |
|---|---|---|---|---|---|
| | | No. | % | No. | % |
| A | 250 | 165 | 66.00 | 85 | 34.00 |
| B | 247 | 147 | 59.51 | 100 | 40.49 |
| C | 63 | 41 | 65.08 | 22 | 34.92 |
| Total | 560 | 353 | 63.04 | 207 | 36.96 |

What draws attention to the data presented is the similarity in the percentage of valid defect reports in the individual buildings. The comparability of the results obtained is the result of the relatively large size of the research sample, which eliminates the divergence of results.

Taking into account the different warranty periods on individual buildings, in order to unify the presentation of statistical characteristics of the testing ground, the occurrence of both the reports of defects as well as their justification was calculated per one month of warranty per one commercial unit. The results of the calculations are presented in Table 3.

**Table 3.** Defect statistics per unit.

| Building | Number of Defects/Month/One Flat | Number of Valid Defects/Month/One Flat | Number of Unfounded Defects/Month/One Flat |
|---|---|---|---|
| A | 0.0543 | 0.0359 | 0.0185 |
| B | 0.0537 | 0.0320 | 0.0217 |
| C | 0.0486 | 0.0316 | 0.0170 |
| Average | 0.0531 | 0.0335 | 0.0196 |

The authors also analyzed the validity of the defects reported within 4 months from the date of acceptance, which is related to the typical time of finishing the premises by the owners (Table 4), and after 12 months, namely, the period when a significant number of premises are already finished and used (Table 5).

**Table 4.** Defect statistics after 3 months from acceptance.

| Building | Number of Defects | Valid | | Unfounded | |
|---|---|---|---|---|---|
| | | No. | % | No. | % |
| A | 61 | 34 | 55.74 | 27 | 44.26 |
| B | 43 | 24 | 55.81 | 19 | 44.19 |
| C | 12 | 9 | 75.00 | 3 | 25.00 |
| Total | 116 | 67 | 57.76 | 49 | 42.24 |

**Table 5.** Defect statistics after 12 months from acceptance.

| Building | Number of Defects | Valid | | Unfounded | |
|---|---|---|---|---|---|
| | | No. | % | No. | % |
| A | 153 | 96 | 62.75 | 57 | 37.25 |
| B | 135 | 79 | 58.51 | 56 | 41.48 |
| C | 46 | 27 | 58.70 | 19 | 41.30 |
| Total | 334 | 202 | 60.48 | 132 | 39.52 |

The analysis performed helped us to formulate two important statements. One is that the owners are very active in making warranty claims in the first 3 months from the date of acceptance. The analysis revealed that 116 reports took place in the first months of use, which accounts for 20.71% of all reports included in the study within only 11% of the study's timeframe. The other observation is the increasing percentage of reported defects, which for the first 3 months was 57.76%, for 12 months 60.48%, and at the end of the study (after half the warranty period) 63.04%. The validity of the defects is presented cumulatively in Figure 2. The deviation of the result for Building C for the first three months of the warranty was due to its smaller size in relation to the other two, which was also associated with a small number of reports, namely, 12 of them.

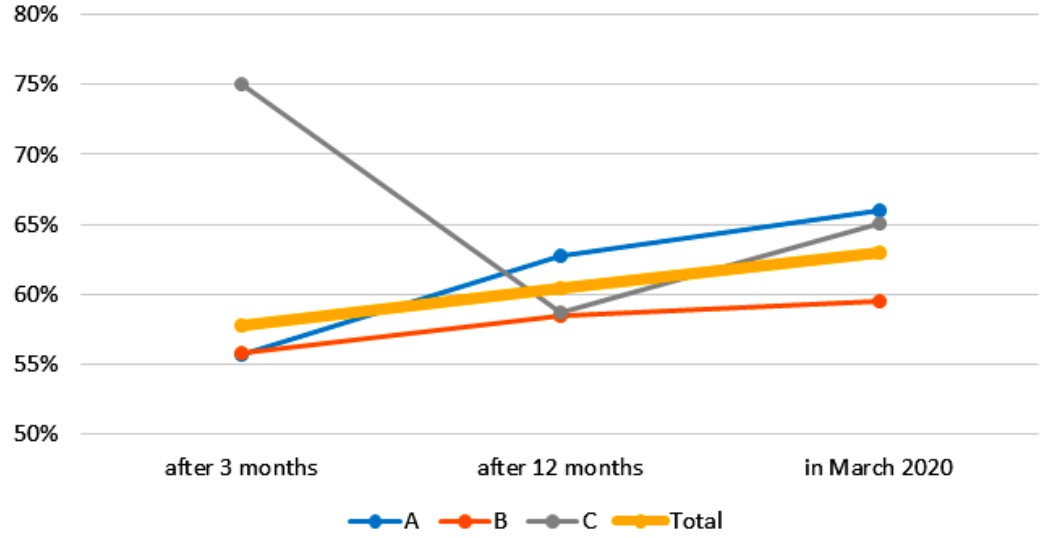

**Figure 2.** Percentage of valid claims under warranty, presented cumulatively.

### 4.3. The Classification of Defects

Based on the analysis of the literature, as well as on the results of the research conducted during the warranty period, the authors propose a three-stage qualification of defects: low, medium, and high significance. The division into these groups corresponds to many publications [3]. In this paper, the following definitions are adopted:

- Defects of high significance—defects causing the appearance of a fault that prevents proper operation of the premises, which may cause a threat to the health or life of people using the property. Removal of very important defects usually requires the elimination of the causes and their effects. Repair works are associated with large financial and material outlays and are time-consuming. Examples of very important defects include flooding of the property, cracking of a structural element, short-circuit of the electrical system, causing fire, or leakage of the roof sheathing.
- Significant defects—faults causing limitations in the proper operation of the premises. This formulation identifies a number of defects that reduce the possibility of unrestricted use.

However, they do not force the cessation of use because they do not constitute a direct threat to the health and life of the users. Due to the large dispersion of the types of these defects, it is not possible to clearly determine the degree of technical and technological complexity of their removal. Examples include the following: intercom failure, decaying tiles on the balcony, unsealing of window and balcony joinery, or slow main door locks.

- Defects of low significance—a minor defect that does not hinder the operation of the premises. This term is used by the authors to describe a defect that has a visual or cosmetic effect, which does not influence the functional properties of the premises. The removal of these defects does not generate the necessity of using any advanced technology, while the time of their removal may vary. They can be illustrated by the following examples: wall scratches, paint chips, spontaneous scratching of the glass, or loosening the silicone next to balcony tiles.

Taking into account the indicated classification, this paper also analyzes the significance of the defects occurring. The results of the significance tests are presented in Table 6.

**Table 6.** Statistics of defect significance.

| Building | Defects of High Significance | Significant Defects | Defects of Low Significance |
|---|---|---|---|
| A | 7.88% | 55.15% | 36.97% |
| B | 19.05% | 51.70% | 29.25% |
| C | 4.88% | 70.73% | 24.39% |
| Average | 12.18% | 55.53% | 32.29% |

As can be seen, the largest percentage, both globally and for individual buildings, was significant defects. It should be noted here that the class of these defects includes issues related to the regulation of woodwork, not the operation of the intercom system or improper connection of the electrical system. The defects of high significance, on the other hand, constitute a state of emergency, therefore in their case, a large discrepancy in frequency can be observed. For Building B, as much as 19.05% were highly significant defects, while Building C had only 4.88% of such reports. The results of the preliminary research also revealed that about 1/3 of the valid reports concerned minor defects, mainly of a cosmetic nature.

## 4.4. Relationship of Reported Defects and the Location

The next steps of the preliminary research were the investigation of the relationship between the reported defects during the warranty and their location. Two cases were considered:

- the defect has been reported by the manager, which is the same as its location in common parts,
- the fault has been reported by the owner of the flat and does not concern elements other than those covered by the boundaries of the flat (additional spaces such as storage rooms or lockers are excluded).

The authors, after analyzing the data from Tables 7 and 8, point out some interesting observations. The first one is the low activity of property managers in the initial phase of the building operation—an average of 9.20% of defects reported during the first 3 months—which stands in contrast to the activities of apartment owners—an average of 22.05% of the reports during the first 3 months from the date of building acceptance. Another observation is the practical equalization of the percentage of reports after 12 months: 57.47–58.95%. On this basis, it can be hypothesized that for the developer in the construction industry, the key period for stabilization of the occurrence of defects is the first 12 months.

**Table 7.** Defects reported by the manager concerning the common parts.

| Building | Number of Defects | After 3 mth | | After 12 mth | |
|---|---|---|---|---|---|
| | | No. | % | No. | % |
| A | 40 | 2 | 5.00 | 24 | 60.00 |
| B | 25 | 1 | 4.00 | 10 | 40.00 |
| C | 22 | 5 | 22.73 | 16 | 72.73 |
| Total | 87 | 8 | 9.20 | 50 | 57.47 |

**Table 8.** Defects reported by the owners of the flat concerning the residential part of the building.

| Building | Number of Defects | After 3 mth | | After 12 mth | |
|---|---|---|---|---|---|
| | | No. | % | No. | % |
| A | 204 | 54 | 26.47 | 120 | 58.82 |
| B | 213 | 40 | 18.78 | 121 | 56.81 |
| C | 41 | 7 | 17.07 | 29 | 70.73 |
| Total | 458 | 101 | 22.05 | 270 | 58.95 |

*4.5. Statistical Analysis of Defect Types*

Taking into account reports appearing during the warranty period, the defects were divided into several groups. The first three groups included reports concerning problems with electrical, plumbing, and central heating (c.h.) installations in apartments. The next groups are reports concerning window and door joinery. Quite often, there were defects consisting of dampness, mainly of walls, from different causes. Separately, reports of wall scratches and other problems related to elements of apartment finishing (such as floors) were grouped. The last group is various defects occurring in the area of balconies and terraces. For each of the buildings, the number of defects was determined in the range of 1–3, 4–6, 7–12, and 13–26 months from the date of acceptance. The list is presented in Table 9.

**Table 9.** The number of defects in groups.

| Building | Time (mth) | Electr. Install. | Plumb. Install. | c.h. Install. | Joinery | Dampness | Scratches | Terraces | Other |
|---|---|---|---|---|---|---|---|---|---|
| A | 1–3 | 26 | 2 | 9 | 12 | 3 | 4 | 5 | 0 |
| | 4–6 | 3 | 0 | 0 | 8 | 0 | 7 | 0 | 1 |
| | 7–12 | 5 | 16 | 5 | 21 | 9 | 3 | 12 | 2 |
| | 13–26 | 11 | 7 | 3 | 25 | 8 | 21 | 19 | 3 |
| Total in A | | 45 | 25 | 17 | 66 | 20 | 35 | 36 | 6 |
| B | 1–3 | 20 | 2 | 3 | 9 | 6 | 0 | 1 | 2 |
| | 4–6 | 9 | 6 | 1 | 6 | 5 | 4 | 12 | 1 |
| | 7–12 | 11 | 4 | 4 | 8 | 5 | 5 | 10 | 1 |
| | 13–26 | 9 | 8 | 7 | 33 | 18 | 15 | 14 | 8 |
| Total in B | | 49 | 20 | 15 | 56 | 34 | 24 | 37 | 12 |
| C | 1–3 | 4 | 2 | 1 | 4 | 0 | 0 | 0 | 1 |
| | 4–6 | 0 | 1 | 1 | 2 | 0 | 2 | 0 | 1 |
| | 7–12 | 2 | 2 | 3 | 4 | 8 | 7 | 1 | 0 |
| | 13–26 | 5 | 1 | 1 | 6 | 0 | 3 | 0 | 1 |
| Total in C | | 11 | 6 | 6 | 16 | 8 | 12 | 1 | 3 |
| Total | | 105 | 51 | 38 | 138 | 62 | 71 | 74 | 21 |

The most numerous group is woodwork defects, and the least numerous group is central heating system faults. These dependencies are the same in all analyzed buildings.

In order to confirm that the number of defects appearing in individual groups is not accidental, the correlation between the numbers of defects in Building A and Building B in the first three months after acceptance was checked (Table 10). Building C was not taken into account due to insufficient data. Then, a correlation was established between all defects reported in all three buildings over the entire observation period (Table 11).

**Table 10.** The number of defects in the groups in the first three months.

| Time [m-th] | Electr. Install. | Plumb. Install. | c.h. Install. | Joinery | Dampness | Scratches | Terraces |
|---|---|---|---|---|---|---|---|
| 1–3 | 26 | 2 | 9 | 12 | 3 | 4 | 5 |
| 4–6 | 20 | 2 | 3 | 9 | 6 | 0 | 1 |

Pearson correlation r = 0.93; significance level $p$ = 0.003.

**Table 11.** The number of defects in the groups in all buildings.

| Building | Electr. Install. | Plumb. Install. | c.h. Install. | Joinery | Dampness | Scratches | Terraces |
|---|---|---|---|---|---|---|---|
| A | 45 | 25 | 17 | 66 | 20 | 35 | 36 |
| B | 49 | 20 | 15 | 56 | 34 | 24 | 37 |
| C | 11 | 6 | 6 | 16 | 8 | 12 | 1 |

Correlations: between A and B, r = 0.86, $p$ = 0.013; between A and C, r = 0.66, $p$ = 0.106; between B and C, r = 0.51, $p$ = 0.24.

The analyses performed allow us to confirm the hypothesis about the lack of randomness of the collected data. Correlation for data from Table 10 allows us to confirm a very strong correlation, while the significance level allows us to reject the hypothesis of zero correlation. For the data in Table 11, the indicators show a very strong correlation between the data for Buildings A and B and a strong correlation for Building C.

For the data concerning all defects (Table 12), the correlation coefficients and the coefficients of determination for individual defect groups were checked. This was to find the relationship between the analyzed groups.

**Table 12.** The number of defects in the group in individual time intervals.

| Time (mth) | Electr. Install. | Plumb. Install. | c.h. Install. | Joinery | Dampness | Scratches | Terraces |
|---|---|---|---|---|---|---|---|
| 1–3 | 50 | 6 | 13 | 25 | 9 | 4 | 6 |
| 4–6 | 12 | 7 | 2 | 16 | 5 | 13 | 12 |
| 7–12 | 18 | 22 | 12 | 33 | 22 | 15 | 23 |
| 13–26 | 25 | 16 | 11 | 64 | 26 | 39 | 33 |

The strongest correlations were found between "joinery" and "scratches" (r = 0.91; $R^2$ = 0.82), "joinery" and "dampness" (r = 0.88; $R^2$ = 0.78), "joinery" and "dampness" (r = 0.88; $R^2$ = 0.78), "scratches" and "terraces" (r = 0.93; $R^2$ = 0.86), "scratches" and "dampness" (r = 0.76; $R^2$ = 0.58), and "dampness" and "terraces" (r = 0.92; $R^2$ = 0.84).

Figures 3–5 show the graphs of the number of defects in individual time intervals, respectively, for defects that appear in a similar distribution over time.

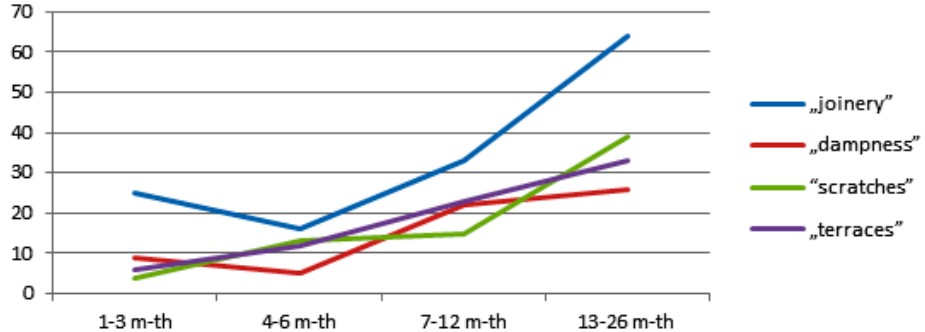

**Figure 3.** The graphs of the distribution of defects: "Joinery", "Dampness", "Scratches", and "Terraces".

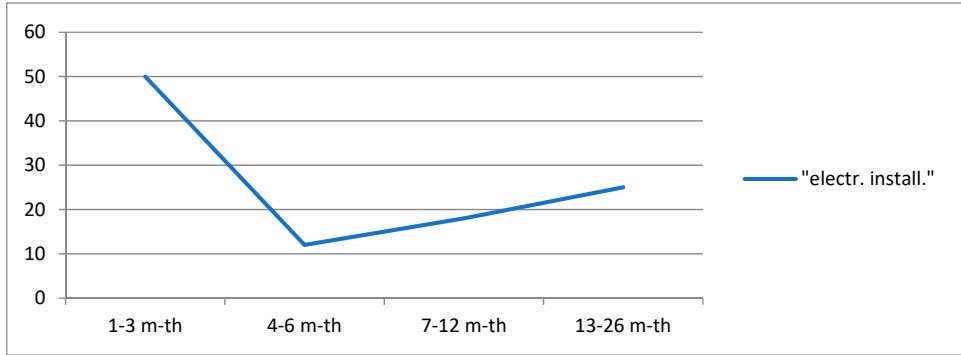

**Figure 4.** The graphs of the distribution of defects: "Electr. Install.".

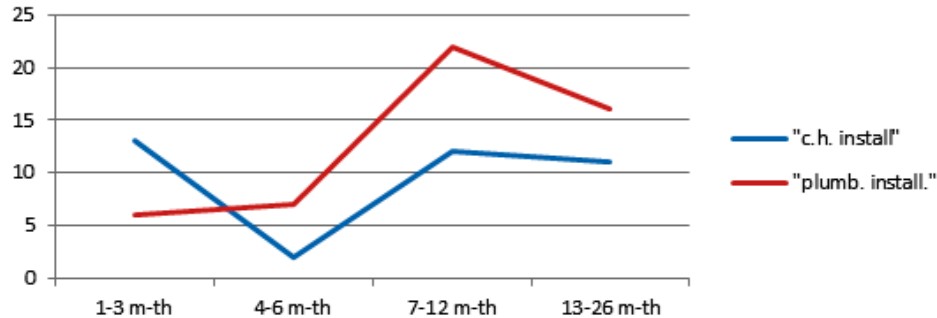

**Figure 5.** The graphs of the distribution of defects: "c.h. Install." and "Plumb. Install.".

Figure 3 illustrates the groups of defects with the highest correlation. These defects are characterized by a relatively small number of cases reported in the first half of the year after the building acceptance and a gradual increase in the subsequent warranty period. The characteristics of defects associated with the electrical installation are definitely different, where by far the largest number of defects are reported in the initial warranty period and their number decreases with time. The pattern of faults in other installations, for which the level of occurrence is similar throughout the observed warranty period, is slightly different.

## 5. Conclusions

A key stage in the mutual relations between the developer and the buyer of an apartment, which, however, does not put an end to the mutual commitment, is the acceptance of works. In the Polish legal environment, the rules of liability are defined by the guarantee and the warranty. The warranty is valid by law and is regulated in detail in the Civil Code.

The paper analyzes warranty notifications in three multifamily residential buildings in the period from their acceptance: January 2018 and August 2018 to March 2020. In total, 560 reports of defects during the warranty were identified. The analyses showed 353 defects qualified as valid and 207 considered as unfounded. Due to the small amount of data, the research results cannot be generalized but allow for the formulation of research hypotheses that will be verified in future studies. The results of the preliminary research reveal that owners are very active in making warranty claims in the first three months from the date of acceptance, and, on the other hand, the percentage of validity of the claims increases with time. In terms of the significance of defects, the largest percentage concerned significant defects. Approximately one-third of reasonable claims involved minor defects, mainly of a cosmetic nature. The results of the preliminary research revealed little activity of property managers in the initial phase of the buildings' operation, which is the opposite of that of apartment owners. Reports of defects in window and door joinery, moisture, wall scratches, and those appearing in the area of balconies and terraces are characterized by a relatively low number of cases reported in the first half of the year after the building is commissioned and a gradual increase in the subsequent warranty

period. On the other hand, faults related to electrical installation are most often reported in the initial period of the warranty, and their number decreases with time.

Understanding defects is a vital prerequisite to their prevention and elimination. So far, the research has focused mainly on defects appearing at the stage of the acceptance of works. The analyses of the defects occurring during the warranty period could fill the research gap, which may significantly affect the development of defect management procedures and the creation of a knowledge map concerning the frequency of defects in particular places of the building and building elements. There are specific costs associated with repairing the defects. Knowledge about those occurring in residential buildings can, therefore, be used for better planning of the investment budget.

The limitation of the research is the small test sample, which only allows for the formulation of research hypotheses concerning the dependence of the appearance of defects during the warranty period. Initially, the research may have had some bias, being of the same builder, a very close period of time, and buildings with similar characteristics. The authors will conduct further research. A larger amount of data will allow us to confirm the conclusions of the present paper.

**Author Contributions:** The individual contribution and responsibilities of the authors were as follows: E.P.: literature review, writing—original draft preparation, writing—review and editing; J.M.: conceptualization, resources, methodology, data curation. All authors have read and agreed to the published version of the manuscript.

**Funding:** This research received no external funding.

**Conflicts of Interest:** The authors declare no conflict of interest.

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
