# Peer review of "Analysis of Defects in Residential Buildings Reported during the Warranty Period"

_applsci, doi:10.3390/app10176123_

Round 1

Reviewer 1 Report

Thank you for your revisions. My concerns in the last version have been revised. 

Author Response

Response to the comments of Reviewer 1

Thank you for your revisions. My concerns in the last version have been revised. 

Thank you very much.

Reviewer 2 Report

Abstract:

The aim of the paper can’t be the analysis itself.

Introduction:

You need to motivate the aim of the paper for an international reader.

In the abstract you claim that you are conducting a qualitative analysis, where is it? The qualitative data is not described in the data collection.

Did you conduct interviews? How many? Where the interviews structured, semi-structured? Where is the interview guide?

Data:

Are the buildings apart of the same project? Is the building produced by the same developer and constructor?

Section 4.4 first sentence rewrite. We do not write about us self like that.

You need to explain what c.h. install. is.

Author Response

Response to the comments of Reviewer 2

The author wish to thank the Reviewer for all comments that helped to enrich and improve the paper. The reviewer’s remarks and requests have been considered carefully by the author. All the requested revisions have been addressed. All the improvements and changes in the manuscript are marked with red colour. The author’ answers are presented in the questions that follow.

- Abstract:

The aim of the paper can’t be the analysis itself.

We changed the aim of the paper (line 10-11).

- Introduction:

You need to motivate the aim of the paper for an international reader.

We tried to motivate the aim of the paper for an international reader (line 54 – 57; 61; 64).

- In the abstract you claim that you are conducting a qualitative analysis, where is it? The qualitative data is not described in the data collection.

Did you conduct interviews? How many? Where the interviews structured, semi-structured? Where is the interview guide?

Information about conducting a qualitative analysis was our mistake. We removed this sentence. We didn’t conduct interviews. The data was collected only through analysis of the reports on the state of defects made by the inspector during the warranty period.

- Data:

Are the buildings apart of the same project? Is the building produced by the same developer and constructor?

We added information: ”The buildings are a part of the various  projects but are produced by the same developer and constructor” (line 200 – 201).

- Section 4.4 first sentence rewrite. We do not write about us self like that.

We rewritten this sentence (line 316-317).

- You need to explain what c.h. install. is.

We explain it (Line 339).

Reviewer 3 Report

The research is interesting and shows the impact of defects in building construction.

Although there are initially related factors, as the authors indicate, there are limitations and they should be taken into account for further research.

As indicated in the review of the state of the art, the authors Zalejska & Hungary, it is important "the relationship between the characteristics of the building, the size of the developer/provider company and the type of defect", so initially the research may have some bias, being the same builder, very close period of time and buildings with similar characteristics.

With respect to the document, in line 72 reference is made to "section 8", so it must be a typo.

Greetings.

Author Response

Response to the comments of Reviewer 3

The author wish to thank the Reviewer for all comments that helped to enrich and improve the paper. The reviewer’s remarks and requests have been considered carefully by the author. All the requested revisions have been addressed. All the improvements and changes in the manuscript are marked with red colour. The author’ answers are presented in the questions that follow.

- The research is interesting and shows the impact of defects in building construction.

Although there are initially related factors, as the authors indicate, there are limitations and they should be taken into account for further research.

In the Conclusions part, we highlighted the limitations of the research.

- As indicated in the review of the state of the art, the authors Zalejska & Hungary, it is important "the relationship between the characteristics of the building, the size of the developer/provider company and the type of defect", so initially the research may have some bias, being the same builder, very close period of time and buildings with similar characteristics.

We added in the Conclusions part, that “initially the research may have some bias, being the same builder, very close period of time and buildings with similar characteristics”.

- With respect to the document, in line 72 reference is made to "section 8", so it must be a typo.

It was our mistake. We removed this sentence.

Round 2

Reviewer 2 Report

I can still not see that the serious data and methodology problems are addressed.

How the article is written is also below what one must be able to expect at this level, this has not improved since the previous revisions.

I have now on two occasions pointed out that the article has serious flaws (data and methodology problems).

As a part of the first revision the authors added in the abstract that they had conducted qualitative studies without any further description.

Author Response

Response to the comments of Reviewer 2

Point 1:

I can still not see that the serious data and methodology problems are addressed.

How the article is written is also below what one must be able to expect at this level, this has not improved since the previous revisions.

I have now on two occasions pointed out that the article has serious flaws (data and methodology problems).

Response 1:

The authors wish to thank the Reviewer for all comments that helped to enrich and improve the paper. While we appreciate the reviewer’s feedback, we respectfully disagree. All the requested by Reviewers revisions have been addressed. Considering the first version of the article, it has been completely changed and improved. The main improvements and changes in the manuscript are marked with red colour. There are as follows:

  • The part “4. Method” is added in article, dedicated to the methodology used in the study.
  • The Abstract and Conclusions sections are changed and significantly expanded.
  • The academic contributions of this study is explained.
  • The article has been extended to include statistical analyzes which prove that the results from the data set allow for the formulation of preliminary conclusions.

We are ready to correct all the shortcomings of the article but we would appreciate if the Reviewer could provide more precise guidance.

Point 2:

As a part of the first revision the authors added in the abstract that they had conducted qualitative studies without any further description.

Response 2:

Information about conducting a qualitative analysis was our mistake. We have removed this sentence from the article.

Round 3

Reviewer 2 Report

You need to add more data, and you need to write about your findings in a way that shows that you understand science and scientific method.

At a bear minimum you need to show the reader how little your data can show us and that the reader should not put very little weight on your findings. You have data on from one developer and one constructor, you have no possibility to do any type of generalization. At a minimum you need to recognize and show that you understand that.

In the first sentence in the new abstract you claim that the aim of the article is to fill research gaps, no is is not. You data and the method you have applied do not have any means filling any research gap, as a minimum you need to show that you understand that.

Author Response

Response to the comments of Reviewer 2

The authors wish to thank the Reviewer for all comments. We hope that the requested revisions have been addressed. The improvements and changes in the manuscript are marked with red colour. The author’ answers are presented in the questions that follow.

Point 1:

You need to add more data, and you need to write about your findings in a way that shows that you understand science and scientific method.

Response 1:

We agree with the Reviewer that we have the small amount of data and the research results cannot be generalized. We indicate that in lines: 11-12; 411 - 412 and 432 – 433.

Point 2:

At a bear minimum you need to show the reader how little your data can show us and that the reader should not put very little weight on your findings. You have data on from one developer and one constructor, you have no possibility to do any type of generalization. At a minimum you need to recognize and show that you understand that.

Response 2:

We indicate that there are preliminary studies that allow only for the formulation of research hypotheses that will be verified in the future studies. We show the reader this in lines: 10-13; 17; 66-70; 313; 316; 411- 413; 432-436.

Point 3:

In the first sentence in the new abstract you claim that the aim of the article is to fill research gaps, no is is not. You data and the method you have applied do not have any means filling any research gap, as a minimum you need to show that you understand that.

Response 3:

We removed this sentence from the article and we changed the aim of the article (lines 10- 13 and 67-70).

Round 4

Reviewer 2 Report

ok

This manuscript is a resubmission of an earlier submission. The following is a list of the peer review reports and author responses from that submission.

Round 1

Reviewer 1 Report

This paper presents an analysis of building defects from the quantity and quality perspectives. Results show that owners are active in making warranty claims in the first three months since commissioning, but the percentage of defects is increased with time. The study is helpful for building defects management.

However, I am concerned about the methods used in the study. From my perspective, this study includes two major methods: data collection and statistical analysis. Both parts of the method are not presented with details and the method section is missing. Can you please add a section to present methods for addressing the proposed problem in this study? In addition, can you please carefully present the academic contributions of this study and summarize the contributions in the Abstract, Discussion and Conclusions sections? Thank you.  

Reviewer 2 Report

Main comments

You need to write a motivation

You write that: The acceptance of construction works is regulated differently in the legal systems of each country. You need to motivate that this article is interesting for an international reader.

An abstract typically include: the purpose or aim of the paper, description of data and data size, discretion of empirical method, description of what makes this article original, the findings and the implications of the findings.

An introduction typically include: a motivation, the purpose or aim, a paragraph that place the article in the literature and explains why it is original, description of data and data size, discretion of empirical method, and a paragraph describing how the rest of the article will be presented.

The amount of data seems to small; this make generalization impossible.

The authors should discuss the implications of the results.

Minor comments

Figure 1 should be made horizontal,

Are you sure that figure 1 should be in the introduction and not in section 3, as well as the explanation of the figure.

I would recommend creating a separate data section.

I would recommend creating a table that describe the data (or the three buildings).

Reference 4 seems to have a strange formatting.

Reference 5, 26 and 27, it is normal to give an English translation in brackets [ translation ].